# Optical Properties of Sc_n_Y_n_ (Y = N, P As) Nanoparticles

**DOI:** 10.3390/nano13182589

**Published:** 2023-09-19

**Authors:** Fotios I. Michos, Alexandros G. Chronis, Michail M. Sigalas

**Affiliations:** Department of Materials Science, University of Patras, GR-26504 Patras, Greece; a.chronis@upatras.gr (A.G.C.); sigalas@upatras.gr (M.M.S.)

**Keywords:** scandium pnictogen, optical properties, DFT, exotic nanoparticles

## Abstract

In this work, using Density Functional Theory (DFT) and Time Dependent DFT, the absorption spectrum, the optical gap, and the binding energy of scandium pnictogen family nanoparticles (NPs) are examined. The calculated structures are created from an initial cubic-like building block of the form Sc_4_Y_4_, where Y = N, P, As after elongation along one and two perpendicular directions. The existence of stable structures over a wide range of morphologies was one of the main findings of this research, and this led to the study of several exotic NPs. The absorption spectrum of all the studied structures is within the visible spectrum, while the optical gap varies between 1.62 and 3 eV. These NPs could be used in the field in photovoltaics (quantum dot sensitized solar cells) and display applications.

## 1. Introduction

Industrial revolution, the lifestyle of the developed world, urbanization, globalization, and technological advancements have contributed to escalated environmental pollution and the depletion of fossil fuel resources [1]. Climate change is one of the most serious issues facing the worldwide community, and it directly affects fuel supply and energy production [2,3]. Increasing emissions of CO_2_ into the atmosphere from human activities, forest fires, global warming, greenhouse gas emissions, and deforestation negatively affect climate change [4,5]. Also, more and more studies have been conducted on how to reduce CO_2_ emissions over both short- and long-term time horizons [6]. Renewable energy consumption is crucial for achieving sustainable environmental goals, and it discourages fossil fuel use in the energy mix, thereby aiming for a sustainable environment. The photovoltaic (PV) industry, solar cells, and other greenhouse technologies have played a significant part in achieving the goal of low carbon development and accomplishing sustainable economic growth [7,8,9].

Nanotechnology, as one of the most developing fields today, offers innovative materials that find application in energy production and conservation. More specifically, semiconductor nanoparticles (NPs), which are commonly named quantum dots (QDs), have great potential for applications in medical diagnostics, drug delivery, gene therapy, medicine, and biology, both in vivo and in vitro, in areas including pharmacokinetics, biosensors, and bio-imaging [10,11,12]. Also, quantum dots are widely used in the field of photovoltaics technologies because they benefit from unique quantum confinement effects [13,14,15].

Quantum dots composed of a combination of at least one of the pnictogen family (phosphorus (P), arsenic (As), and nitrogen (N)) are widely used in applications based on their optical and structural properties. More specifically, nitrogen quantum dots have unique and distinct photoluminescence properties, with an increasing percentage of nitrogen compared to the neighboring carbon dots [16], while doping of N is a promising strategy to modulate electronic, chemical, and structural functionalities of graphene (G) and graphene quantum dots (GQDs) in energy and environmental applications, such as supercapacitors, batteries, sensors, fuel cells, solar cells, and photocatalysts [17,18]. Also, there have been many scientific studies to explore synthetic methods (such as ultrasonic and electrochemical exfoliation, solvothermal treatment, blender breaking, milling crushing, and pulsed laser irradiation), properties, and modifications of black phosphorus quantum dots (BPQDs), exhibiting a broad range of applications in the fields of bioimaging, fluorescence sensing, nonlinear optical absorbers, cancer therapy, intelligent electronics, photovoltaics, optoelectronics, and flexible devices [19,20]. Recent research has shown that Sc doping in Titanium Dioxide (TiO_2_) can be used for applications in photocatalytic water-splitting technology in low-cost and eco-friendly hydrogen production [21], while the interstitial dimers of Sc formed in silicon can introduce several intermediate-bands (IBs) in the band gap, making the material a high-efficiency IB solar cell [22]. Furthermore, the conduction band structure and electron mobility in rocksalt ScN were recently studied using DFT [23], while there has been progress to overcome the materials engineering challenges to grow high-quality epitaxial, nominally single crystallin metal/semiconductor superlattices based on transition metal nitrides [24,25].

In the present study, using the Density Functional Theory and Time Dependent DFT, the absorption spectrum, the optical gap, and the binding energy of scandium–pnictogen family nanoparticles are examined. The calculated structures are of the form Sc_x_Y_x_ where Y = N, As, P, while additionally a wide range of stable configurations are presented. This work aims to provide non-toxic and easy to find semiconducting NPs, which can be used in the field of renewable energy source such as photovoltaic technologies [26,27].

## 2. Materials and Methods

In this manuscript, the optical and the structural properties of more than thirty different geometries were calculated by Density Functional Theory and Time Dependent DFT (TD-DFT). The structures studied herein were initially formed after elongation of an initial geometry in one and two perpendicular directions. From the study of structurally stable configurations, stable morphologies were formed. Using the gradient corrected functional of Perdew, Burke, and Ernzerhof (PBE) [28], with the triple-ζ quality def2-TZVP basis set [29], the geometries of the studied structures were optimized. Subsequently, with the help of the TD-DFT on the ground state geometries, the lowest singlet vertical excitation energies and the ultraviolet-visible absorption spectra were calculated. Note that the initial calculations were made only for the small nanostructures, due to the lack of experimental data on the absorption spectrum of the large structures. We were very concerned about choosing the functional for each group of the scandium–pnictogen family. It is true that if we use a consistent functional (especially PBE) for all studied absorption spectra, the calculations would be faster (and perhaps more comparable to each other). However, the results from the comparison with the EOM-CCSD [30] functional showed us that it is better and more accurate to calculate the absorption spectrum with a different functional for each group of the scandium–pnictogen family. The functionals used to calculate the absorption spectra were PBE, M06-2X [31], and CAM-B3LYP [32] for Sc_x_N_x_, Sc_x_P_x_, and Sc_x_As_x_ nanoparticles, respectively. A specified script representing each excitation as a Gaussian function centered on the TD-DFT calculated excitation energy and with a standard deviation (broadening) of 0.1 eV was used to calculate the absorption spectrum. Also, in order to approximate the peak shape, which is observed experimentally, the latter value of the spectrum was chosen. Additionally, the calculation package TURBOMOLE [33] was used in all calculations, while the Particle Swarm Optimization (PSO) [34] was used to ensure that the studied nanostructures are indeed realistic.

## 3. Results

Classified as a rare-earth element, scandium has attracted much research interest in its properties. The optical, structural, and electronic properties of Sc, such as the refractory nitrides ScN, make it an excellent candidate semiconductor for optoelectronic and DMS applications [35,36]. Also, Yu Gong and Mingfei Zhou showed that the cyclic Sc(μ-N)2Sc molecules dimerize on annealing to form a cubic Sc_4_N_4_ cluster with tetrahedral symmetry, which is a fundamental building block for ScN NPs and crystals [37]. Note that in small-size nanocrystals, the Sc_4_N_4_ units prefer to arrange into the rectangular-like wire structures, and they arrange into the compact rectangular-like or cubic-like configurations in large-size nanocrystals [38]. Recent research has shown that the cage unit tends to arrange into the compact configurations, and the occupied positions of N atoms shift from the surface towards the center of the coordination site with the increasing number of Sc atoms, while there is a study in which the calculated optical spectra (by using DFT) suggest that 2D tetragonal ScN and YN nanosheets have high visible light absorption efficiency [39,40].

All of the studied structures were created from an initial cubic-like building block of the form Sc_4_Y_4_, where Y = N, P, As. Figure 1 shows the elongation of this block, from which new stable structures with interesting optical properties could be created. So, the nanostructures of Sc, which are formed, fall into three categories. In the first category belong the elongated nanoparticles in one direction (1D); in the second belong the NPs in two vertical directions; and, in the third category, belong the exotic structures. Regarding the shape of the structure (Figure 1), we notice that the geometry of Sc_4_N_4_ NP is almost a perfect cube, because the corresponding angles are almost 90 degrees, while the geometry of phosphorus and arsenic structures is like a distorted cube (the angles formed are acute and obtuse). Also, the third category of studied NPs is formed by creating a hole in the structures (categories one and two) after extracting atoms from them. Note that the study of examined nanostructures showed very interesting results in green technology applications.

### 3.1. Sc_x_Y_x_ NPs

Figure 2 shows the absorption spectra of the initial cubic-like building blocks for the studied NPs, i.e., the structures Sc_4_Y_4_, where Y = N, P, As. In the case of Sc_4_N_4_, the first five excitation energies on the AS appear at 2.59, 3.63, 3.82, 4.07, and 4.44 eV, with corresponding absorption of 0.10, 0.13, 0.01, 0.20, and 0.02. Similarly, the first five excitation energies in the case of Sc_4_P_4_ appear at 2.79, 2.93, 3.11, 3.27, and 3.42 eV, with the corresponding absorption varying around 0.007, 0.01, 0.09, 0.24, and 0.01. When Sc_4_As_4_ is considered in the calculations, the first five excitation energies of the AS appear at 2.44, 2.56, 2.63, 2.83, and 2.99 eV, with corresponding absorbance of 0.01, 0.04, 0.05, 0.06, and 0.02. Also, as we can observe, the smallest optical gap is 2.44 eV, and it appears in the case of Sc_4_As_4_, while for the structures Sc_8_Y_8_ where Y = N, P, As, the smallest optical gap shifts to lower energies (Figure 3). From the comparison of Figure 2 and Figure 3, we observed that in the case of Sc_8_N_8_, the optical gap appeared at 2.41 eV, i.e., 0.18 eV lower than the initial cubic structure and the optical gap of Sc_8_P_8_ appears at 2.43 eV, i.e., 0.50 eV lower than the corresponding Sc_4_P_4_, while the value of the optical gap for Sc_8_As_8_ appears at 2.21 eV and it is lower by 0.23 eV than the corresponding cubic structure. Furthermore, the strength of the absorbance of the Sc_8_Y_8_ structures shows an upward trend as we move into larger energies. The first five excitation energies for the Sc_8_N_8_ on the AS appear at 2.41, 2.53, 2.74, 2.99, and 3.14 eV, while the corresponding absorbances are about 0.03, 0.05, 0.02, 0.06, and 0.03. For the Sc_8_P_8_ NPs, the first five excitation energies appear at 2.43, 2.61, 2.80, 3.02, and 3.10 eV and the corresponding absorbances are about 0.01, 0.15, 0.01, 0.08, and 0.09. Similarly, in the case of Sc_8_As_8_, the first five peak positions appear at 2.21, 2.39, 2.52, 2.81, and 2.96 eV with absorbance of 0.05, 0.11, 0.04, and 0.09, respectively.

The following is the stability of the examined NPs by the binding energies per formula unit (BE per f.u.). The BE, E_B_, is defined as follows:$$E_{B}\left({Sc}_{x}Y_{x}\right)=xE\left(Sc\right)+xE\left(Y\right)-E({Sc}_{x}Y_{X})$$
where E(Sc) is the energy of the isolated atoms of Sc, E(Y) is the energy of the isolated atoms of Y = N, P, As, and the index x indicates each examined formula of the structure (formula unit). E(Sc_x_Y_x_) is the total energy of the examined NPs. Also, the binding energy per f.u., (E_b_) is
$$E_{b}\left({Sc}_{x}Y_{x}\right)=\frac{E_{BE}\left({Sc}_{x}Y_{X}\right)}{x}=E\left(Sc\right)+E(Y)-E({Sc}_{x}Y_{X})/x$$

The BE per f.u. for the Sc_4_N_4_, Sc_4_P_4_, and Sc_4_As_4_ is 10.45, 8.53, and 7.40 eV, respectively.

#### 3.1.1. Elongated Sc_x_Y_x_ NPs along One Direction

As mentioned above, the scandium NPs are divided into three categories. First, nanostructures were created by elongation of the initial cubic-like building block, Sc_4_Y_4_ (Y = N, P, As), along one direction. The general form of the created NPs is Sc_x_Y_x_, where x = 8, 16, 24. As we can observe from Figure 4, the structure with x = 8 consists of two initial cubic-like building blocks, the structure with x = 16 consists of four corresponding cubes, and the structure with x = 24 contains six individual initial corresponding cubes. These structures are optimized as geometrically stable due to not finding imaginary frequencies. We clarify that the Sc_x_N_x_ and Sc_x_As_x_ nanostructures are almost the same as the Sc_x_P_x_ shown.

The results of calculating optical properties of the examined NPs and BE per f.u are shown in the Table 1. We observe that as the number of atoms in the structure increases, a significant decrease in the optical gap is observed in three cases of Sc_x_Y_x_ in this category. For the Sc_x_N_x_ NPs, the optical gap decreases from 2.59 eV (for Sc_4_N_4_) gradually to 2.41 and 2.29 eV (for Sc_8_N_8_ and Sc_16_N_16_, respectively), while for Sc_24_N_24_, it is 2 eV. The BE per f.u. has particularly high values compared to Μg_x_Y_x_ NPs [41], which indicates that the Sc monopnictogen structures are significantly more stable. Also, the binding energy is 10.45 eV for Sc_4_N_4_, while it increases to 11.57 eV for the Sc_24_N_24_. In the case of Sc_x_P_x_ NPs, it is noted that for x = 4, the optical gap is 2.93 eV, while as the nanostructure becomes bigger, a large reduction in the OG is observed. More specifically, the OG for Sc_8_P_8_ is 0.5 eV less than Sc_4_P_4_, while it is 2.18 eV for the Sc_24_P_24_. In the case of Sc_x_As_x_ NPs in this category, the absorption spectrum and the oscillator strength have the same intensity as the corresponding Sc_x_P_x_. So, the OG is 2.44 and 2.00 eV for Sc_4_As_4_ and Sc_24_As_24_, respectively, and the oscillator strength is 1.18.

#### 3.1.2. Elongated NPs along Two Perpendicular Directions

The second group of Sc nanostructures examined in this work contains the initial building block, which was elongated along two perpendicular directions. The general form of the proposed nanoparticles is Sc_x_N_x_, where x = 16, 24, 32, 36, and 48. The resulting nanostructures are shown in Figure 5 and the calculated optical properties are collected in Table 2. Note that the nanostructures depicted in the first row of Figure 5 were derived from Sc_8_Y_8_ and they have a square shape, while the corresponding structures in the second row were derived from Sc_16_Y_16_ and they have a rectangular shape. The optical gap for the examined Sc_x_N_x_ in this category ranges between 2.33 eV and 2.07 eV, while the Sc_36_N_36_ and Sc_48_N_48_ nanostructures for which we do not give a value for the optical gap are unstable. The BE per f.u. for the examined NPs is approximately the same as those of NPs elongated along one direction and range between 11.68 eV and 12.13 eV for Sc_16_N_16_ and Sc_48_N_48_, respectively.

When the second part of the NP is phosphorus, the optical gap is 2.16 eV and 1.73 eV for Sc_16_P_16_ and Sc_48_P_48_ NPs, respectively, while the OS for Sc_36_P_36_ is much higher than other examined nanostructures in this category. Regarding the BE per f.u. for the studied NPs, the calculated values range between 9.91 eV and 10.41 eV. Also, the Sc_36_P_36_ NPs seem to be more stable than Sc_32_P_32_ NPs (with a hole in the center of their geometry) because they have a higher BE by 0.35 eV.

The calculations continued by considering arsenic as the second part of the examined NPs. The BE per f.u. of the smallest nanostructure (Sc_16_As_16_) elongated along two directions is 2.02 eV, while the oscillator strength ranges between 0.02 and 0.10.

A very interesting case from the nanostructures that are examined and created by elongation of the initial building block along one and two perpendicular directions is the one where x = 16. In Figure 6, the absorption spectra of these cases of Sc_16_Y_16_ NPs are shown. A notable general observation is that the most excitation energies are within the visible spectrum. More specifically, for the elongated along one direction Sc_16_N_16_ NP, the first six excitation energies appear at 2.03, 2.29, 2.49, 2.58, 2.73, and 2.87 eV, while for the elongated along two perpendicular directions Sc_16_N_16_ NP, the relative excitation energies are 2.33, 2.41, 2.53, 2.70, 2.93, and 3.09 eV. Also, in the case of Sc_16_P_16_ NP elongated along one direction, the absorption spectrum showed the first six excitation energies at 2.26, 2.36, 2.67, 2.80, 3.01, and 3.09 eV, while for the corresponding NPs elongated along two directions, the AS showed the corresponding energies at 2.16, 2.23, 2.51, 2.84, 3.09, and 3.24 eV. For Sc_16_As_16_ NPs, the calculations showed the first six resonances at 2.07, 2.24, 2.52, 2.80, 2.96, and 3.23 eV for those created from elongation along one direction and 2.02, 2.19, 2.30, 2.60, 2.84, and 3.04 eV for the NPs that were created by elongation of the initial building block along two perpendicular directions.

The calculated binding energy per f.u. versus the number of the formula (x) of each examined NP is shown in Figure 7. It is observed that the nanostructures that present higher BE per f.u. are those that have a lighter atom in the second part of the NP. It takes higher values for the case of nitrogen and a further decrease for the cases of phosphorus and arsenic. So, when the second part of the studied NP is N, P, and As, the BE per f.u. ranges between 10.48 and 12.16 eV, 8.5 and 10.38 eV, and 7.25 and 8.78 eV, respectively. Furthermore, for all cases of Sc_x_Y_x_ (where Y = N, P, As), the corresponding curves have the same trend, i.e., when the formula number is less than 16, the BE per f.u. raises slightly, while when the x is greater than 36, the BE per f.u. is almost unchanged. Also, the flattening of the curve (in the case that x > 36) strongly indicates that the calculated values for the BE per f.u. would not differentiate significantly with the values of the bulk.

### 3.2. Sc_x_Y_x_ Exotic NPs

The third category of scandium nanostructures studied in this work includes the exotic NPs, which are so named because of their morphology characteristics. The stability of these structures was checked by calculating their vibrational spectrum. No negative frequencies were found for the 24 different NPs shown. The general form of those structures is Sc_x_Y_x_, where Y = N, P, As and x ranges between 12 and 40. The studied exotic NPs are divided into two groups. The first group of exotic NPs includes structures whose morphology does not change (or only slightly changes) with their geometry optimization (Figure 8), while the second group includes nanostructures that change their shape (Figure 9).

As we can observe from Figure 8, NPs have several familiar shapes. So, there are NPs with L-type (a, b c cases), with C-type (d case), with Z-type (e), with Cross-type (h, I, k), with W-type (g), with O-type (n), and with D-type (o) shapes. Also, the (j) nanostructure consists of three crosses joined together (without a cube in the center for each cross), the (l) nanostructure forms a 2D-cross, and the (m) one consists of two joined crosses. The last three structures that did not change their morphology with geometry optimization are the (p), (q), and (r) cases. The first one consists of two individual squares of 2 × 2 initial cubic-like building blocks, the second is formed by three identical squares, and the last one comes from the previous structure, but by removing two cubes from the perimeter (it looks like two letters ‘’Γs’’ are formed anti-diametrically). The initial configurations of the structures shown in Figure 8h,l are similar, except that the outer-left and out-right atoms (four on each side) have been removed in Figure 8h’s case. After geometry optimization, the final configuration shown in Figure 8l has been almost the same as its initial configuration. However, the final configuration shown in Figure 8h has been significantly changed from its original configuration. This is a clear indication that the stable configurations consist of cubic-like building blocks Sc_4_Y_4_ (Y = N, P, As).

The optical gap (OG), the oscillator strength (OS), and the BE per f.u. of the proposed optimized structures were numerically examined. The results of this set of calculations are shown in Table 3. Note that the structures, in which the OG has not been calculated, are not stable. When the second part of the studied NPs in this category is nitrogen, the optical gap takes various values between 1.65 and 2.34 eV, while the BE per f.u. varies from 11.30 eV to 11.86 eV. As we can observe, the structure of this category that has the minimum optical gap (1.65 eV) with the best resonance (0.09) and BE per atom (11.86 eV) is the Sc_40_N_40_ NP (case q). When the second part of the exotic NPs is the phosphorus, the calculated OG is 1.95, 1.89, 1.88, 1.84, 1.82, 1.76, 1.74, and 1.72 eV, while the resonance is 0.05, 0.06, 0.12, 0.01, 0.10, 0.09, 0.05, and 0.14 for the cases (e), (l), (g), (f), (p), (o), (m), and (r), respectively. Also, the BE per f.u range is between 9.40 eV and 10.12 eV. In the case of arsenic, the optical gap is 1.98, 1.90, 1.89, 1.89, 1.84, 1.82, 1.81, 1.80, 1.75, and 1.73 eV for the cases (h), (i), (a), (c), (d), (b), (f), (n), (l), (p), (e), (g), and (r), respectively. Similarly to the aforementioned group of nanostructures, Sc_40_N_40_ NP has the largest BE per atom (8.68 eV) and the strongest resonance (0.24), while its OG is 1.65 eV. Notable cases of this category are also (m) and (q) cases, in which the OG is 1.63 and 1.68 eV, respectively.

The second group of studied exotic NPs includes nanostructures that change their morphology after geometry optimization. Figure 9 shows all of the examined structures created again from the initial cubic-like building block and presents the initial and final nanostructures. It is noteworthy that the final morphology changes significantly when the second element is N, while the structures that have in the second element P or As the morphology remain almost unchanged. The final shape that each optimized NP is strongly related to is the initial shape before optimization. Also, for the T-type shapes, and for the C-type with the extensions (cases (a), (b), (c), and (d) of the relevant Figures), the final shape of the central building block of the optimized NP takes a hexagonal form. In Figure 9e,f, NPs with A-type and cross-H-type shapes are presented, respectively. It is observed that all optimized A-type NPs are similar to each other, while the cross-H-type Sc_x_As_x_ NP is not stable and it is not presented.

The optical gap (OG), the oscillator strength, and the binding energy per f.u. of the proposed structures were also numerically examined. The results of this set of calculations are collected in Table 4. When the second part of the studied NPs in this category is nitrogen, the optical gap varies from 2.13 eV to 2.47 eV and the oscillator strength is weak, while the values for the BE per atom vary from 11.37 to 11.71 eV. Also, when the second part of the exotic NPs is the phosphorus, the calculated OG is 1.88, 1.85, and 1.76 eV, while the BE per f.u. is 9.75, 9.70, and 9.96 eV for the cases (d), (c), and (e), respectively. In the case of arsenic, the calculated OG is 2.09, 2.00, 1.94, 1.94, and 1.73 eV for the cases (a), (e), (b), (d), and (c), respectively. The smallest calculated value for the BE per f.u. is 5.14 eV and the larger is the 8.57 eV.

Therefore, among the most characteristic NPs studied is Sc_16_Y_16_ (where Y = N, P, As), and the corresponding absorption spectra are shown in the following graphs. In Figure 10, an L-type Sc_12_Y_12_ and Sc_16_Y_16_ composed after elongation of the initial building block along one and two perpendicular directions are additionally shown. A notable general observation is that the exotic examined NPs have much weaker oscillator strengths than the elongated corresponding NPs. However, in the case of nitrogen, the absorption spectrum has an increasing trend, and it was calculated at 2.12, 2.22, 2.23, 2.29, and 2.42 eV for Z-type, Middle-type, L-type, Cross-type, and T-type NPs. The same behavior is observed regarding the case of phosphorus and arsenic. The optical gap takes values of 1.95 eV for Z-type, 2.06 for L-type, 2.08 for Middle-type, 2.11 for T-type, and 2.52 eV for Cross-type Sc_16_P_16_ NPs, while the OG was calculated at 1.80, 1.89, 2.11, 2.20, and 2.27 eV for Z-type, Middle-L-type, Τ-type, L-type, and Cross-type of Sc_16_As_16_ NPs, respectively.

## 4. Conclusions

The structural, electronic, and optical properties of Sc_x_Y_x_ NPs (where x = 4 up to 48 and Y = N, P As) were examined in this work. Using Particle Swarm Optimization (*PSO*), it was confirmed that for the case where x = 4, the structure with the lowest energy is the cubic-like building block. A total of 34 different nanostructures were studied and created by elongation of the initial cubic-like building block, Sc_4_Y_4_ (Y = N, P, As), along one, two, and three perpendicular directions. Also, geometry optimizations and structural stability for all studied NPs were carried out with Density Functional Theory.

The absorbance spectrum was calculated with Time Dependent Density Functional Theory for Sc_x_Y_x_ (where x = 4,8,16 and Y = N, P As), while for larger NPs, only the optical gap was calculated. The functionals used to calculate the absorption spectra were PBE, M06-2X, and CAM-B3LYP for Sc_x_N_x_, Sc_x_P_x_, and Sc_x_As_x_ NPs, respectively. The small differences in the BE per f.u. of the initial cubic-like building block and the studied NPs indicate that the good structural stability of the proposed morphologies is based on the good structural stability of the initial building block. It is also worth noting that the optical gap (OG) generally decreases with increasing nanoparticle size. For studied NPs created by elongation along one direction, the OG starts from 2.59, 2.93, and 2.44 eV to 2.00, 2.18, and 2.00 eV in the case of nitrogen, phosphorus, and arsenic, respectively. Similarly, in the case of Sc_x_Y_x_ NPs elongated in two directions, the OG starts (when x = 16) from 2.33, 2.16, and 2.02 eV to (when x = 48) 2.07, 1.73, and 1.77 eV for Y = N, P, and As, respectively. Also, regarding the exotic NPs, whether their morphology changes or not with the geometry optimization, the OG decreases as the number of atoms in the structure increases. In all three groups of examined NPs, the exotic nanostructures show the OG at lower energies than the corresponding structures elongated in one and two directions, while in all studied cases, the OG ranges from 1.62 (for the Sc_40_As_40_ NP) to 3.6 eV. Also, for all examined Sc_x_Y_x_ NPs, the Z-type NP has the lowest optical gap.

In conclusion, the studied Sc_x_Y_x_ NPs show their optical gap within the visible spectrum and indeed at quite low frequencies, depending on the morphology and material. The lowest optical gap found is 1.62 eV (while the absorbance is 0.09) for the case of Sc_40_P_40_ NP, which consists of three joined initial cubic-like building blocks. The absorption spectrum of all the structures studied indeed covers most of the visible spectrum. Such structures can find applications in a variety of fields that require absorption in the visible spectrum, as in sensors, catalysis, photocatalysis, in biolabeling, in LEDs, and in emerging low-cost third-generation solar cells technology (quantum dots sensitized solar cell). The results of this work could be utilized by theorists and experimentalists in further research on these very interesting and powerful sectors.

## Figures and Tables

**Figure 1 nanomaterials-13-02589-f001:**
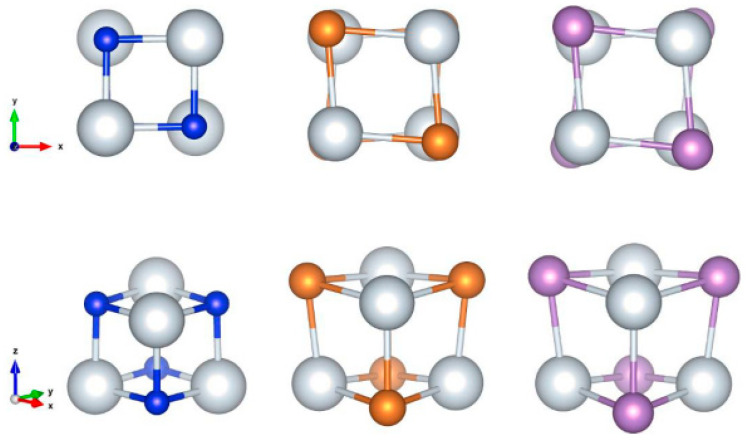
The initial cubic-like building block Sc_4_Y_4_ (Y = N, P, As). The top view refers to the top view of the cubes, while the bottom view refers to the side view (45 degree). The white color corresponds to the Sc atom, the blue one to the N atom, the orange one to the P atom, and the purple one to the As atom.

**Figure 2 nanomaterials-13-02589-f002:**
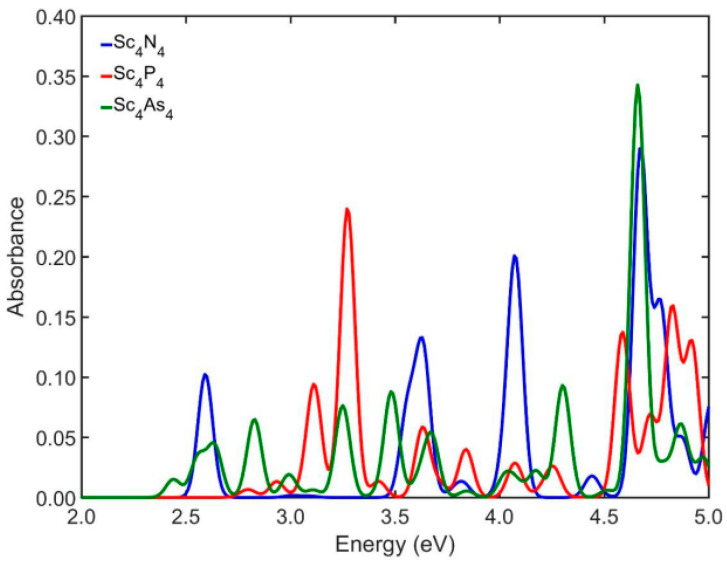
The UV/Vis absorbance spectra of the initial cubic-like building blocks of Sc_4_Y_4_ (Y = N, P, As) NPs.

**Figure 3 nanomaterials-13-02589-f003:**
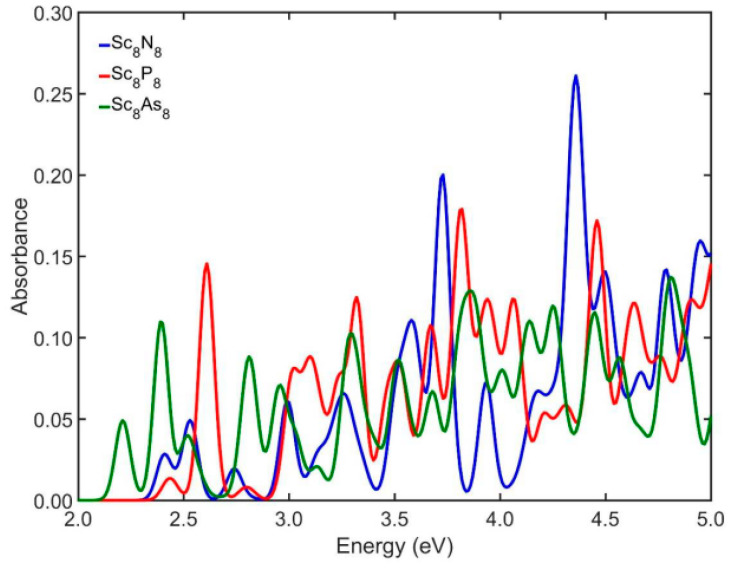
The UV/Vis absorbance spectra of the initial cubic-like building blocks of Sc_8_Y_8_ (Y = N, P, As) NPs.

**Figure 4 nanomaterials-13-02589-f004:**
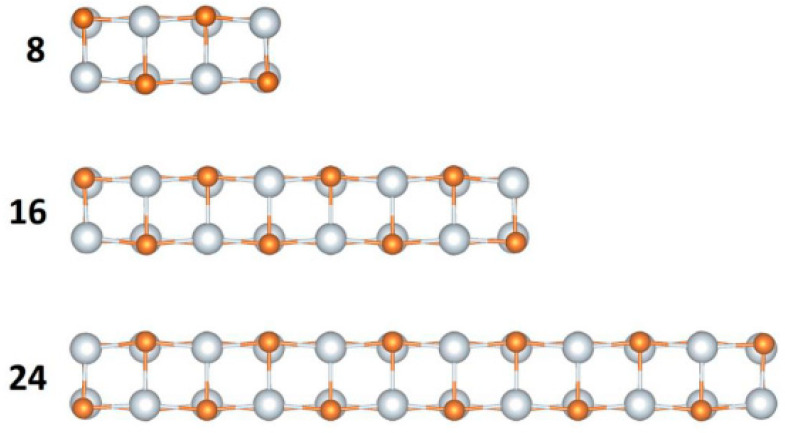
The examined NPs (Sc_x_Y_x_, where x = 8, 16, 24), which were created after elongation of the initial cubic-like building block Sc_4_Y_4_ (Y = P, N, As) along one direction.

**Figure 5 nanomaterials-13-02589-f005:**
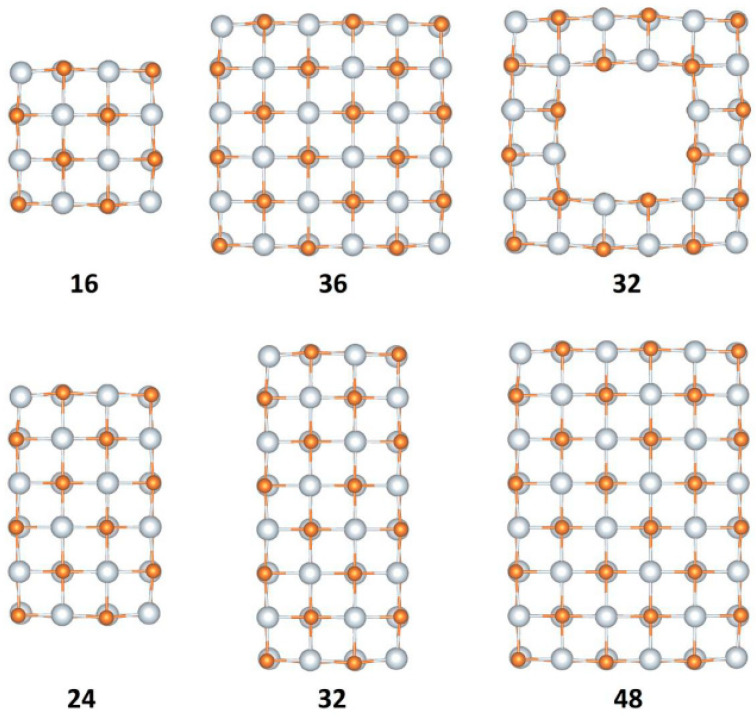
The examined NPs (Sc_x_Y_x_, where x = 16, 24, 32, 36, and 48), which were created after elongation of the initial cubic-like building block Sc_4_Y_4_ (Y = P, N, As) along two perpendicular directions.

**Figure 6 nanomaterials-13-02589-f006:**
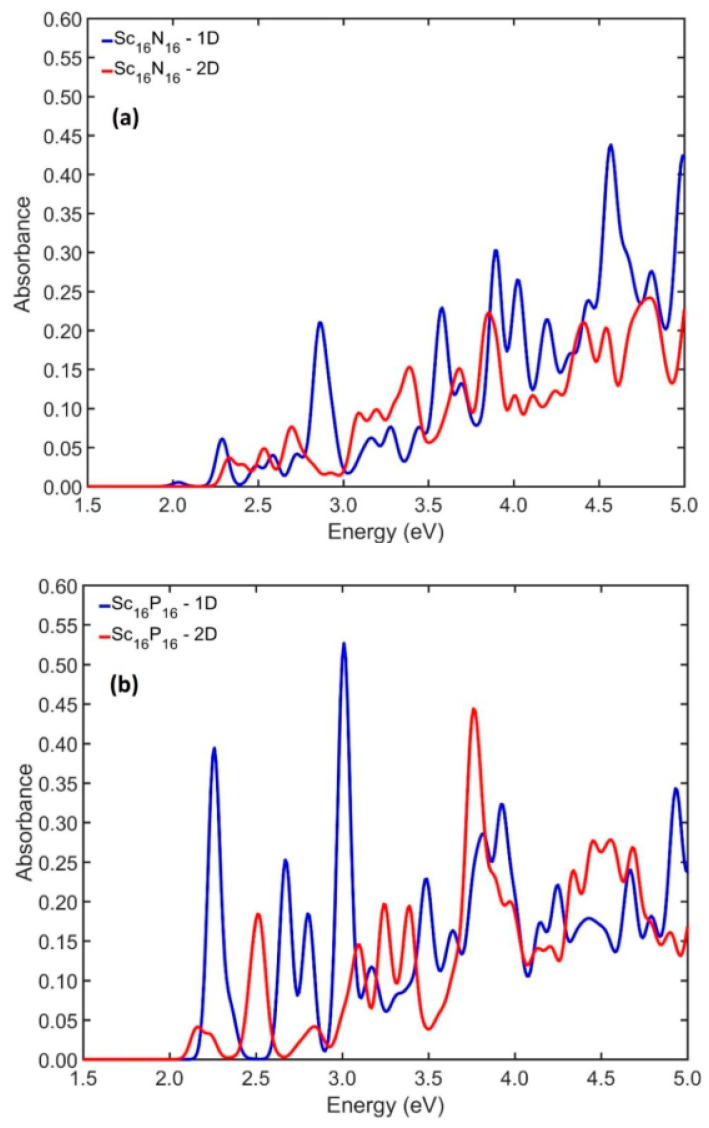
The UV–visible spectra of Sc_16_Y_16_ (Y = N (**a**), P (**b**), As (**c**)) when considered after elongation along one direction (blue line) and two perpendicular directions (red line).

**Figure 7 nanomaterials-13-02589-f007:**
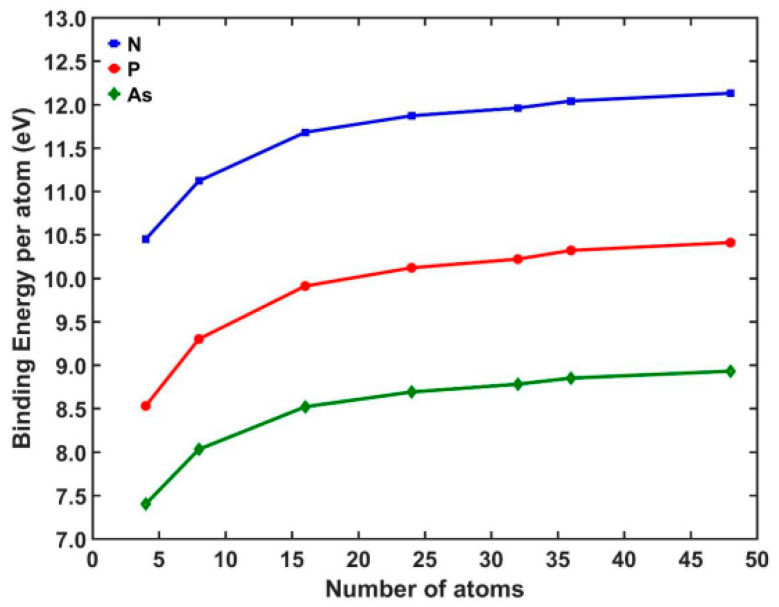
The binding energy per f.u. (Eb) of the Sc_x_Y_x_ NPs for all of the examined cases.

**Figure 8 nanomaterials-13-02589-f008:**
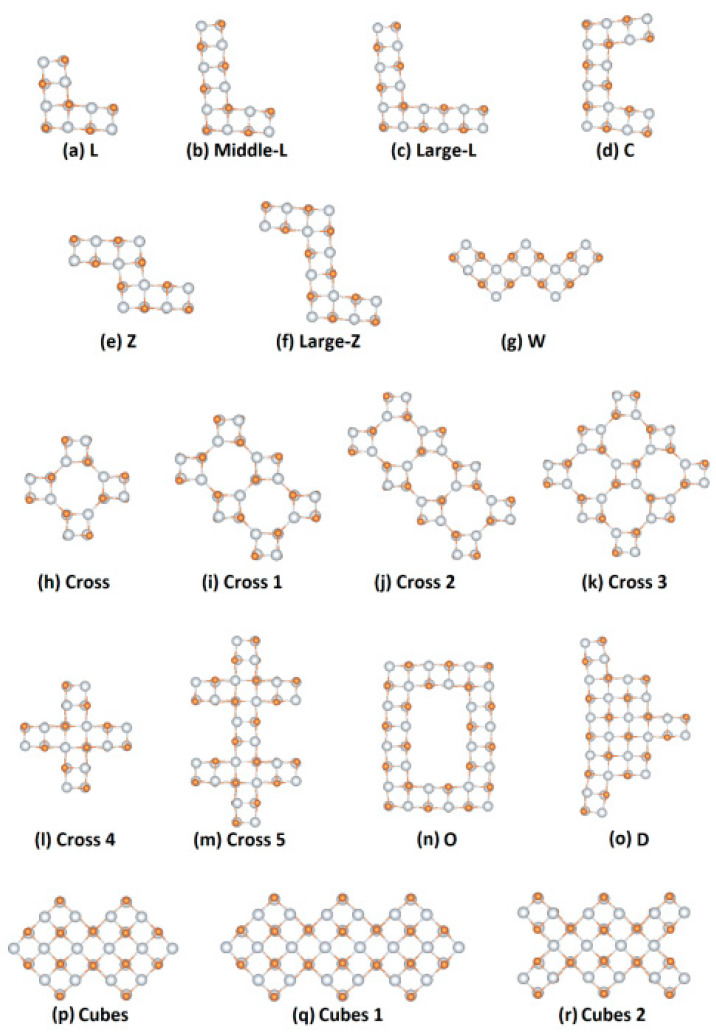
Exotic nanostructures with familiar shapes and their morphologies do not change (or only slightly change) with their geometry optimization (Sc_x_Y_x_, where Y = N, P, As, and x varies from 12 to 40). Each morphology is the same for all the combinations of elements (N, P, As) studied.

**Figure 9 nanomaterials-13-02589-f009:**
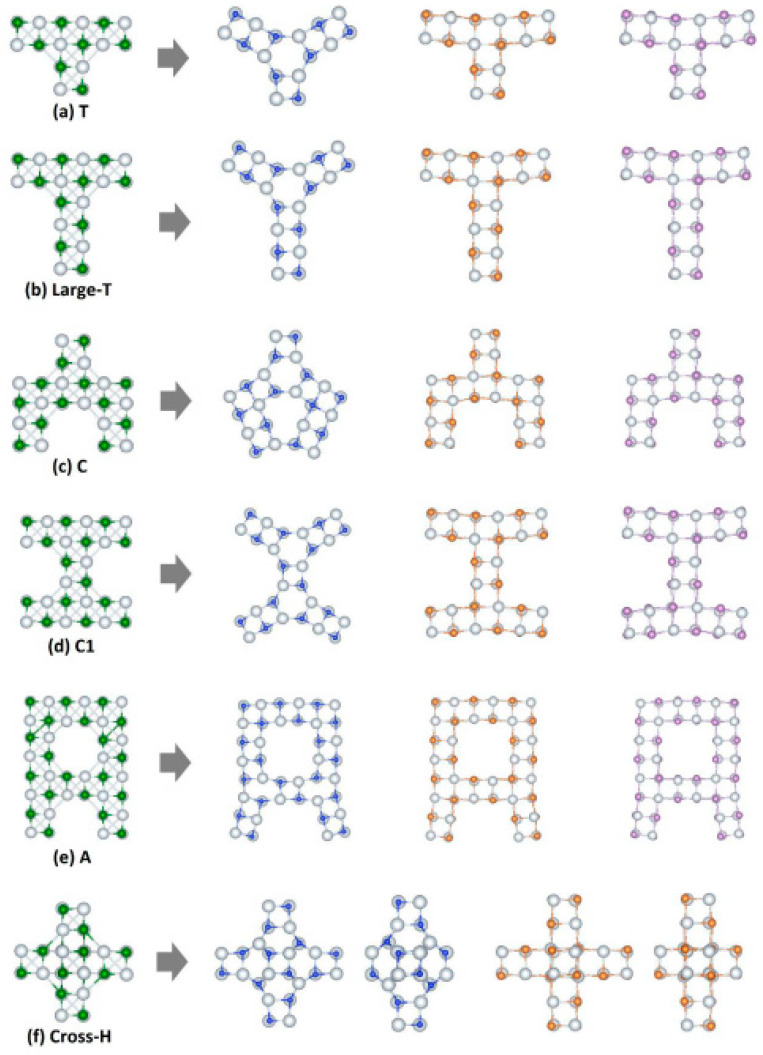
Exotic nanostructures with familiar shapes and their morphologies change with their geometry optimization (Sc_x_Y_x_, where Y = N, P, As, and x varies from 16 to 40). The structures with a green color are before geometry optimization, while the rest of the structures are optimized. The blue, brown, and purple atoms refer to N, P, and As, respectively.

**Figure 10 nanomaterials-13-02589-f010:**
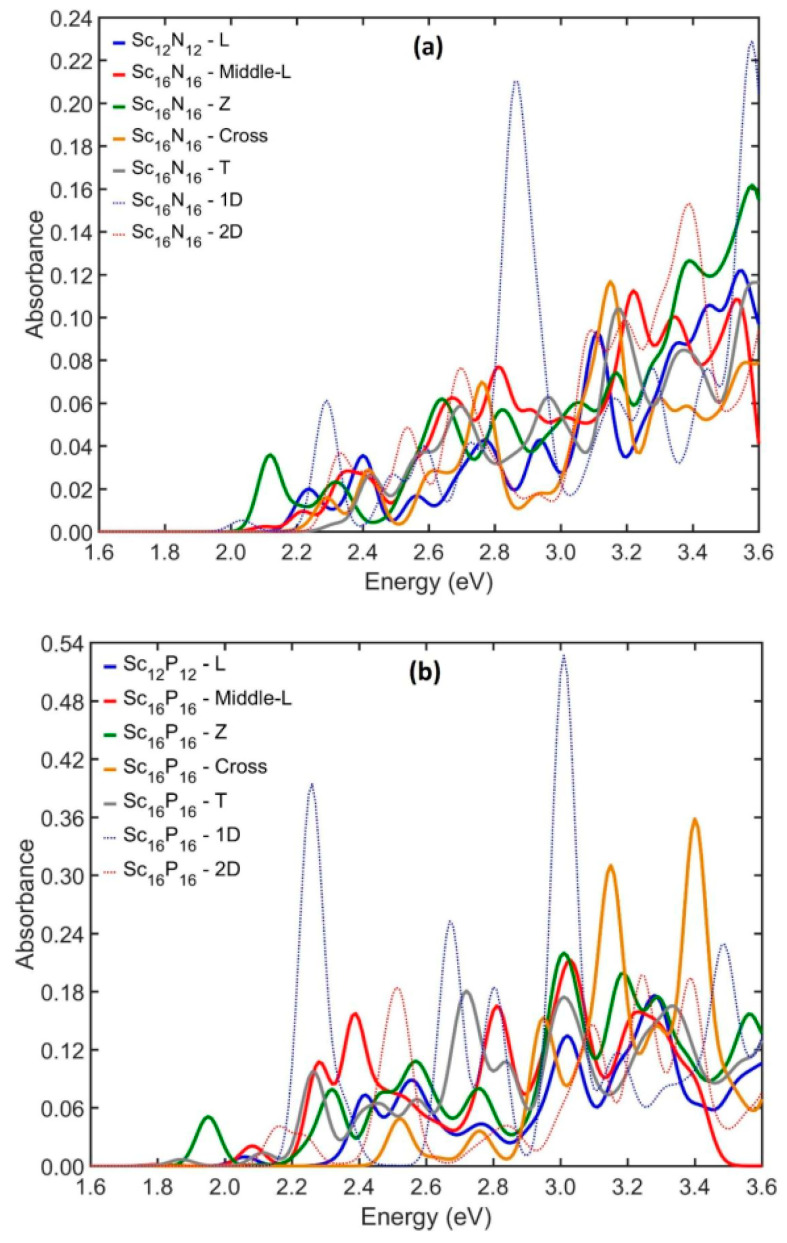
The UV–visible spectra of exotic NPs ScxYx with familiar shapes (where (**a**) Y = N, (**b**) Y = P, (**c**) Y = As and x = 12 and 16) and examined Sc_16_Y_16_ (where (**a**) Y = N, (**b**) Y = P, (**c**) Y = As) created after elongation of the initial cubic-like building block.

**Table 1 nanomaterials-13-02589-t001:** Binding energy (BE) per f.u., optical gap (OG), and oscillator strength (OS) for Sc_x_Y_x_ NPs (x = 4, 4, 16 and Y = N, P, As), which were created after elongation of the initial cubic-like building block Sc_4_Y_4_ (Y = N, P, As) along one direction.

	NP	OG (eV)	OS	BE per Atom (eV)
**N**	Sc_4_N_4_	2.59	0.10	10.45
	Sc_8_N_8_	2.41	0.03	11.12
	Sc_16_N_16_	2.29	0.06	11.46
	Sc_24_N_24_	2.00	0.003	11.57
**P**	Sc_4_P_4_	2.93	0.01	8.53
	Sc_8_P_8_	2.43	0.01	9.30
	Sc_16_P_16_	2.26	0.39	9.68
	Sc_24_P_24_	2.18	0.92	9.80
**As**	Sc_4_As_4_	2.44	0.02	7.40
	Sc_8_As_8_	2.21	0.05	8.03
	Sc_16_As_16_	2.07	0.47	8.36
	Sc_24_As_24_	2.00	1.18	8.47

**Table 2 nanomaterials-13-02589-t002:** Binding energy (BE) per f.u., optical gap (OG), and oscillator strength (OS) for Sc_x_Y_x_ NPs (x = 16, 24, 32, 36, 18 and Y = N, P, As), which were created after elongation of the initial cubic-like building block Sc_4_Y_4_ (Y = N, P, As) along two directions. Style T refers to square-shaped structures in the first raw in Figure 8, and style-O refers to rectangular-shaped structures in the second raw in Figure 8.

	Style	NP	OG (eV)	OS	BE per Atom (eV)
**N**	T	Sc_16_N_16_	2.33	0.04	11.68
		Sc_36_N_36_	-	-	12.04
		Sc_32_N_32_	2.14	0.02	11.72
	O	Sc_24_N_24_	2.22	0.02	11.87
		Sc_32_N_32_	2.07	0.04	11.96
		Sc_48_N_48_	-	-	12.13
**P**	T	Sc_16_P_16_	2.16	0.04	9.91
		Sc_36_P_36_	1.89	0.06	10.32
		Sc_32_P_32_	2.18	0.23	9.97
	O	Sc_24_P_24_	1.93	0.04	10.12
		Sc_32_P_32_	1.80	0.08	10.22
		Sc_48_P_48_	1.73	0.02	10.41
**As**	T	Sc_16_As_16_	2.02	0.02	8.52
		Sc_36_As_36_	1.88	0.10	8.85
		Sc_32_As_32_	2.01	0.22	8.59
	O	Sc_24_As_24_	1.83	0.03	8.69
		Sc_32_P_32_	1.71	0.05	8.78
		Sc_48_P_48_	1.77	0.05	8.93

**Table 3 nanomaterials-13-02589-t003:** Binding energy (BE) per f.u., optical gap (OG), and oscillator strength (OS) for exotic optimized Sc_x_Y_x_ NPs (x ranges between 12 and 40 and Y = N, As, P), whose morphologies do not change (or only slightly change) with their geometry optimization. Cases and shapes are shown in Figure 8.

	Case	Shape	NP	OG (eV)	OS	BE per Atom (eV)
**N**	a	L	Sc_12_N_12_	2.25	0.01	11.30
	b	Middle-L	Sc_16_N_16_	2.22	0.01	11.43
	c	Large-L	Sc_20_N_20_	2.32	0.01	11.50
	d	C	Sc2_0_N_20_	2.24	0.01	11.48
	e	Z	Sc1_6_N_16_	2.12	0.04	11.39
	f	Large-Z	Sc_20_N_20_	2.15	0.01	11.48
	g	W	Sc_20_N_20_	1.99	0.06	11.44
	h	Cross	Sc_16_N_16_	2.29	0.02	11.32
	i	Cross 1	Sc_24_N_24_	2.21	0.01	11.47
	j	Cross 2	Sc_32_N_32_	2.33	0.04	11.54
	k	Cross 3	Sc_36_N_36_	2.34	0.02	11.62
	l	Cross 4	Sc_20_N_20_	-	-	11.37
	m	Cross 5	Sc_36_N_36_	-	-	11.48
	n	O	Sc_40_N_40_	2.18	0.01	11.74
	o	D	Sc_36_N_36_	-	-	11.78
	p	Cubes	Sc_28_N_28_	1.87	0.04	11.81
	q	Cubes 1	Sc_40_N_40_	1.65	0.09	11.86
	r	Cubes 2	Sc_32_N_32_	1.94	0.04	11.60
**P**	a	L	Sc_12_P_12_	2.06	0.01	9.49
	b	Middle-L	Sc_16_P_16_	2.08	0.02	9.64
	c	Large-L	Sc_20_P_20_	2.25	0.02	9.72
	d	C	Sc_20_P_20_	2.09	0.04	9.69
	e	Z	Sc_16_P_16_	1.95	0.05	9.60
	f	Large-Z	Sc_20_P_20_	1.84	0.01	9.69
	g	W	Sc_20_P_20_	1.88	0.12	9.65
	h	Cross	Sc_16_P_16_	2.52	0.05	9.40
	i	Cross 1	Sc_24_P_24_	2.48	0.08	9.55
	j	Cross 2	Sc_32_P_32_	2.47	0.15	9.62
	k	Cross 3	Sc_36_P_36_	2.45	0.02	9.69
	l	Cross 4	Sc_20_P_20_	1.89	0.06	9.60
	m	Cross 5	Sc_36_P_36_	1.74	0.05	9.72
	n	O	Sc_40_P_40_	2.04	0.02	9.99
	o	D	Sc_36_P_36_	1.76	0.09	10.03
	p	Cubes	Sc_28_P_28_	1.82	0.10	10.06
	q	Cubes 1	Sc_40_P_40_	1.62	0.21	10.12
	r	Cubes 2	Sc_32_P_32_	1.72	0.14	9.85
**As**	a	L	Sc_12_N_12_	2.18	0.02	8.19
	b	Middle-L	Sc_16_N_16_	1.89	0.01	8.32
	c	Large-L	Sc_20_N_20_	1.98	0.04	8.39
	d	C	Sc_20_N_20_	1.90	0.02	8.35
	e	Z	Sc1_6_N_16_	1.80	0.03	8.27
	f	Large-Z	Sc_20_N_20_	1.89	0.03	8.35
	g	W	Sc_20_N_20_	1.75	0.09	8.32
	h	Cross	Sc_16_N_16_	2.27	0.05	8.13
	i	Cross 1	Sc_24_N_24_	2.26	0.01	8.25
	j	Cross 2	Sc_32_N_32_	-	-	8.31
	k	Cross 3	Sc_36_N_36_	-	-	8.37
	l	Cross 4	Sc_20_N_20_	1.82	0.04	8.26
	m	Cross 5	Sc_36_N_36_	1.63	0.03	8.36
	n	O	Sc_40_N_40_	1.84	0.02	8.61
	o	D	Sc_36_N_36_	1.68	0.04	8.62
	p	Cubes	Sc_28_N_28_	1.81	0.11	8.64
	q	Cubes 1	Sc_40_N_40_	1.65	0.24	8.68
	r	Cubes 2	Sc_32_N_32_	1.73	0.13	8.46

**Table 4 nanomaterials-13-02589-t004:** Binding energy (BE) per f.u., optical gap (OG), and oscillator strength (OS) for exotic optimized Sc_x_Y_x_ NPs (x ranges between 12 and 40 and Y = N, As, P), whose morphologies change with their geometry optimization. Cases and shapes refer to Figure 9.

	Case	Shape	NP	OG (eV)	OS	BE per Atom (eV)
**N**	a	T	Sc_16_N_16_	2.42	0.02	11.43
	b	Large-T	Sc_20_N_20_	2.37	0.01	11.50
	c	C	Sc_24_N_24_	2.40	0.01	11.66
	d	C1	Sc_28_N_28_	2.36	0.01	11.57
	e	A	Sc_40_N_40_	2.13	0.01	11.71
	f	Cross-H	Sc_24_N_24_	2.47	0.01	11.37
**P**	a	T	Sc_16_N_16_	2.11	0.01	9.59
	b	Large-T	Sc_20_N_20_	2.12	0.06	9.69
	c	C	Sc_24_N_24_	1.85	0.05	9.70
	d	C1	Sc_28_N_28_	1.88	0.01	9.75
	e	A	Sc_40_N_40_	1.76	0.01	9.96
	f	Cross-H	Sc_24_N_24_	2.07	0.04	9.50
**As**	a	T	Sc_16_N_16_	2.09	0.07	8.27
	b	Large-T	Sc_20_N_20_	1.94	0.02	8.35
	c	C	Sc_24_N_24_	1.73	0.03	8.35
	d	C1	Sc_28_N_28_	1.94	0.07	8.39
	e	A	Sc_40_N_40_	2.00	0.20	8.57
	f	Cross-H	Sc_24_N_24_	-	-	8.14

## Data Availability

Not applicable.

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
