# Peer review of "Optical Properties of ScnYn (Y = N, P As) Nanoparticles"

_nanomaterials, 2023, doi:10.3390/nano13182589_

Round 1

Reviewer 1 Report

The manuscript of F. I. Michos et al. describes theoretical study of ScxYx NPs. It was predicted that they should show an optical gap within the visible spectrum and indeed at quite low frequencies, depending on the morphology and material. For the readers it will be interesting to know that the lowest optical gap found is 1.62 eV (while the absorbance is 0.09) for the case of Sc40P40 NP, which consists of three joined initial cubic-like building blocks. In total, the absorption spectra of all the structures studied indeed cover most of the visible spectrum. Such structures can be targeting structures for a broad audience in a variety of fields that require absorption in visible spectrum, as in sensors, catalysis, photocatalysis, at biolabeling, in LEDs and in emerging low-cost third-generation solar cells technology (Quantum Dots Sensitized Solar Cell). In view of this, this manuscript can be accepted for publication in Nanomaterials. I suggest the authors to improve the work by addressing the following comment.

1. Please for the most interesting case Sc40P40 NP try to add other pnictogen atoms at symmetrical positions. It is important to know an influence of this substitution on the optical gap.

The manuscript needs final English language polishing before being publishable.

Author Response

First of all, we thank the Reviewer for his/her encouraging and critical comments.

Comment 1

‘’ Please for the most interesting case Sc40P40 NP try to add other pnictogen atoms at symmetrical positions. It is important to know an influence of this substitution on the optical gap.’’

Response

We absolutely agree with the Reviewer and this idea is very interesting. Adding other pnictogen atoms at symmetrical positions in the case of Sc40P40 NP or substitution the scandium with magnesium or titanium or similar elements, is worth considering in the near future when more computation time is available.

Comment 2

‘’The manuscript needs final English language polishing before being publishable.’’

 Response

The Reviewer is absolutely right. There were grammatical errors, which were corrected in the revised manuscript.

Reviewer 2 Report

I have three main problems with this manuscript on ScY nanoparticles: first, it is not put into the context of available knowledge on these systems, second, your methodological approach seems very arbitrary, and third, a large part of the manuscript deals with very unrealistic geometries. On top of that, it consists to a large extent of uncommented numbers, while general conclusions that would lead to physical insight are very sparse. I will discuss these points in more detail below, but as it is, the manuscript should not be published.

To my first main problem: you discuss references that are of no relevance whatsoever to what is the topic here, while you neglect to give an overview about what is already known about the issues you treat here. For instance the very first paragraph: yes, also for me the human influence on the world climate is a very, very important political issue, but this manuscript is definitely not the place to discuss these points. Erase it completely. Perhaps you see yourself that it is not wise to allude to such specific historic events in a text that should stand for eternity, when with your Greek background this year's August is arguably even more relevant. But what will be in 2024? Also the first paragraph under "3. Results": apart from Ref. 35, this is just not relevant.

On the other hand, there are things you should discuss to render a service to the readers (who otherwise have to look for such things themselves): you should state (with references of course) that your systems in the bulk state indeed have the rocksalt structure, of which your Sc4Y4 cube can be seen as the smallest building block. Further, at least for ScN the existence of this building block as molecule has experimentally indeed been established (10.1021/jp070816n), and also the stable structures for given size have been theoretically predicted (10.1016/j.cplett.2012.09.029). Essentially, you do not motivate at all which structures you study here.

You write twice about "the pnictogen-nitrogen family". Conventionally, nitrogen is included in the pnictogens, and conversely, in the second instance you should mention scandium.

To the second main problem: To me, Figs. 2-4 are very ambiguous. I do not think that they can be used to claim that one functional is better than the others. According to which criteria have you come to your decisions? What is the mentioned 8% difference? If at all, I would say that the only consistent way would be to do the indefinite integral of the absorbance functions and compute the squared differences to your EOM-CCSD (if it is indeed accepted that this functional in general has the best predictive capability). But in any case, unless you can give a reasoning why different systems should be best described by different functionals, it would be much, much better to use a consistent functional for all your calculations, thus making them comparable.

You write "NPs that do not have an optical gap means they are not stable". This is very misleading -- probably you want to say that the structures for which you do not give a gap are unstable, but it looks like you imply that if they have no gap, they are unstable.

And the third main problem: I cannot imagine that for instance the structure in Fig. 11h is stable. 10.1016/j.cplett.2012.09.029 have shown that it is essentially the number of nearest-neighbour bonds that determines the stability. And the cross structure from Fig. 11 (with a binding energy of 11.32 eV per atom) can directly transform to the Sc16N16 square structure of Fig. 8 (with a binding energy of 11.68 eV per atom) just by rotating the four 8-atom cubes in alternating directions. Indeed, it looks to me like you did your geometrical optimizations in the ideal symmetry, because in all your figures I never see anything like a spontaneous symmetry breaking. Is this correct? But in any case, at one point you claim that you did look for imaginary frequencies. If this is true, then you should find all unstable configurations. But I doubt that, because computing all the oscillation frequencies for the larger ones among your structures becomes computationally really hard, and perhaps even impossible, because numerical inaccuracies will destroy the positive-definiteness of the dynamical matrix.

Minor editing of English language required

Author Response

First of all, we thank the Reviewer for his/her comments that helped us clarify several aspects in the manuscript and improve its quality significantly.

Comment B1

‘’ To my first main problem: you discuss references that are of no relevance whatsoever to what is the topic here, while you neglect to give an overview about what is already known about the issues you treat here. For instance, the very first paragraph: yes, also for me the human influence on the world climate is a very, very important political issue, but this manuscript is definitely not the place to discuss these points. Erase it completely. Perhaps you see yourself that it is not wise to allude to such specific historic events in a text that should stand for eternity, when with your Greek background this year's August is arguably even more relevant. But what will be in 2024? Also, the first paragraph under "3. Results": apart from Ref. 35, this is just not relevant.’’

Response

We totally agree with the Reviewer. We have deleted the first paragraph under ‘’1. Introduction’’ and replaced it with a new paragraph that is mush more relevant (along with its references) to the studied topic. Also, we replaced the first paragraph under "3. Results" with a new one. The changes to the paragraphs are shown below.

Change

-1. Introduction

Industrial revolution, lifestyle of the developed world, urbanization, globalization and the technological advancements are contributed to escalated environmental pollution and depletion of fossil fuel resources [1]. Climate change is one of the most serious issues facing the worldwide communication and directly affects fuel supply and energy production [2,3]. Increasing emissions of CO2 into the atmosphere from human activities, forest fires, global warming, greenhouse gas emissions and deforestation negatively affect climate change [4,5]. Also, more and more studies have been conducted on how to reduce CO2 emissions over both short- and long-term time horizons [6]. Renewable energy consumption is crucial for achieving sustainable environmental goals and discourages fossil fuel use in the energy mix, aiming for a sustainable environment. The photovoltaic (PV) industry, solar cells and other greenhouse technologies have played a significant part in the achieving the goal of low carbon development and accomplishing sustainable economic growth [7,8,9].

-3. Results

Classified as a rare-earth element, scandium has attracted much research interest in its properties. The optical, structural and electronic properties of Sc, such as the refractory nitrides ScN, make it an excellent candidate semiconductor for optoelectronic and DMS applications [35]. Also, Yu Gong and Mingfei Zhou showed that the cyclic Sc(μ-N)2Sc molecules dimerize on annealing to form a cubic Sc4N4 cluster with tetrahedral symmetry, which is a fundamental building block for ScN NPs and crystals [36]. Noted that in small-size nanocrystals, the Sc4N4 units prefer to arrange into the rectangular-like wire structures and they arrange into the compact rectangular-like or cubic-like configurations in large-size nanocrystals [37]. Recent research has shown that the cage unit tends to arrange into the compact configurations, and the occupied positions of N atoms shift from the surface towards the center of coordination site with the increasing number of Sc atoms, while there is a study in which calculated optical spectra (by using DFT) suggest that 2D tetragonal ScN and YN nanosheets have high visible light absorption efficiency [38, 39].

Comment B2

’On the other hand, there are things you should discuss to render a service to the readers (who otherwise have to look for such things themselves): you should state (with references of course) that your systems in the bulk state indeed have the rocksalt structure, of which your Sc4Y4 cube can be seen as the smallest building block. Further, at least for ScN the existence of this building block as molecule has experimentally indeed been established (10.1021/jp070816n), and also the stable structures for given size have been theoretically predicted (10.1016/j.cplett.2012.09.029). Essentially, you do not motivate at all which structures you study here’’

Response

We add the following references in the first paragraph under ‘’3.Results’’ in order to help readers better understand the studied nanostructures.

  1. Gong, Y., Zhao, & Zhou, M. Formation and characterization of the tetranuclear scandium nitride: Sc4N4. The Journal of Physical Chemistry A, 2007, 111(28), 6204-6207. doi.org/10.1021/jp070816n
  2. Xu, M., Zhang, Y., Zhang, J., Qian, B., Lu, J., Zhang, Y., ... & Chen, X. Structures and properties evolution with size of ScN nanocrystals: A first-principles study. Chemical Physics Letters, 2012, 551, 126-129. doi.org/10.1016/j.cplett.2012.09.029
  3. Li, CG., Zhou, JC., Hu, YF. et al. Computational Studies on the ScnNm (n + m=10) Clusters: Structure, Electronic and Vibrational Properties. J Clust Sci, 2018, 29, 459–468. https://doi.org/10.1007/s10876-018-1352-z
  4. Liu, J., Li, X. B., Zhang, H., Yin, W. J., Zhang, H. B., Peng, P., & Liu, L. M., Electronic structures and optical properties of two-dimensional ScN and YN nanosheets. Journal of Applied Physics, 2014, 115(9). https://doi.org/10.1063/1.4867515

Comment B3

You write twice about "the pnictogen-nitrogen family". Conventionally, nitrogen is included in the pnictogens, and conversely, in the second instance you should mention scandium.’’

Response

We change ‘’the pnictogen-nitrogen family’’ with ‘’scandium-pnictogen family’’ in the four following sentences.

Change

-page 1, Abstract, keywords

-page 1, keywords

-page 1, third paragraph

- page 2, last paragraph of the introduction

Comment B4

‘’To the second main problem: To me, Figs. 2-4 are very ambiguous. I do not think that they can be used to claim that one functional is better than the others. According to which criteria have you come to your decisions? What is the mentioned 8% difference? If at all, I would say that the only consistent way would be to do the indefinite integral of the absorbance functions and compute the squared differences to your EOM-CCSD (if it is indeed accepted that this functional in general has the best predictive capability). But in any case, unless you can give a reasoning why different systems should be best described by different functionals, it would be much, much better to use a consistent functional for all your calculations, thus making them comparable.’’

Response

We were very concerned about choosing the functional to make the best comparison with the more accurate EOM-CCSD functional. It is true that if we use a consistent functional for all studied absorption spectra, the calculations would be much faster (and perhaps more comparable to each other). However, the results from the comparison with the EOM-CCSD functional showed us that it is better and more accurate to calculate the absorption spectrum with a different functional for each group of scandium-prictogen family. For example, in Figure 3, the most suitable functional is M06-2X, since the first three peaks are almost the same as that of EOM-CCSD. Also, the results of PBE functional used for the calculations of absorption spectrum in Fig.2 are very different from the corresponding EOM-CCSD. For the above reason, we decided to calculate the absorption spectra for each group of scandium-pnictogen family with the functional that gave results closer than the corresponding EOM-CCSD functional.

Regarding the ‘’8% difference’’, as shown in Figure 2, PBE and B3LYP functionals are the best in comparison with the EOM_CCSD results.  The optical gap calculated with ΕΟΜ_CCDD is, indeed, almost the same as the corresponding one calculated with the B3LYP functional, while it differs by 8% from the corresponding gap calculated with PBE functional.  However, as we also notice from the Fig.2, the excitation energies calculated with PBE functional (compared to those calculated with B3LYP functional) are in better agreement with the corresponding energies calculated with EOM_CCDT. In conclusion, for this reason the PBE functional is chosen for the approximation and calculation of the absorption spectrum. Also, if the Reviewer considers that the above comment will help the understanding of the manuscript, we can add it to the revised text.

Comment B5

‘’You write "NPs that do not have an optical gap means they are not stable". This is very misleading -- probably you want to say that the structures for which you do not give a gap are unstable, but it looks like you imply that if they have no gap, they are unstable.’’

Response

We change ‘’ NPs that do not have an optical gap means they are not stable’’ with         ’’ the Sc36N36 and Sc48N48 nanostructures for which we do not give a value for optical gap are unstable’’

Change

-penultimate sentence of page 9

Comment B6

‘’And the third main problem: I cannot imagine that for instance the structure in Fig. 11h is stable. 10.1016/j.cplett.2012.09.029 have shown that it is essentially the number of nearest-neighbour bonds that determines the stability. And the cross structure from Fig. 11 (with a binding energy of 11.32 eV per atom) can directly transform to the Sc16N16 square structure of Fig. 8 (with a binding energy of 11.68 eV per atom) just by rotating the four 8-atom cubes in alternating directions. Indeed, it looks to me like you did your geometrical optimizations in the ideal symmetry, because in all your figures I never see anything like a spontaneous symmetry breaking. Is this correct? But in any case, at one point you claim that you did look for imaginary frequencies. If this is true, then you should find all unstable configurations. But I doubt that, because computing all the oscillation frequencies for the larger ones among your structures becomes computationally really hard, and perhaps even impossible, because numerical inaccuracies will destroy the positive-definiteness of the dynamical matrix.’’

Response

The initial configurations of the structures shown in Fig. 11h and 11l are similar except that the outer-left and out-right atoms (4 on each side) have been removed in Fig. 11h case. After geometry optimization, the final configuration shown in Fig. 11l has been almost the same as its initial configuration. However, the final configuration shown in Fig. 11h has been significantly changed from its original configuration. This is a clear indication that the stable configurations consist of cubic-like building blocks Sc4Y4 (Y=N, P, As). (We add the aforementioned comment to the second paragraph under ’’3.2. ScxYx exotic NPs’’). Also, it is mentioned that all the structures are stable with no imaginary frequency except a few as we mentioned in our reply of the previous comment.

Reviewer 3 Report

Using quantum chemistry calculation, this work very extensively characterized structural and optical properties of ScxYx (Y= N, P As) type of nanoparticles (NPs), the studied NPs vary from very small to very large size, and wide variety of morphologies were taken into account. This is an interesting and valuable theoretical work, which has guiding significance for the practical application of ScnYn type quantum dots in the future.

This work may be acceptable after a minor revision:
It is stated "as shown in the Figure 2 for the case Sc4P4...", however caption of Figure 2 indicates that the figure was plotted for Sc4N4. This inconsistency should be revised. In addition, in this figure, it can be seen that the B3LYP peak matches well with the EOM-CCSD peak at ~2.8 eV, while PBE underestimates excitation energy of this peak evidently. I really don't know how the authors drew the conclusion that PBE is the most suitable functional for estimating absorption spectrum of ScxNx. This point should be better clarified.

Author Response

First of all, we would like to thank the referee for the positive evaluation of our work.

Comment 1

‘It is stated "as shown in the Figure 2 for the case Sc4P4...", however caption of Figure 2 indicates that the figure was plotted for Sc4N4. This inconsistency should be revised.’’

 Response

The Reviewer is absolutely right. We change the ‘‘as shown in the Figure 2 for the case Sc4P4, the most suitable functional’’ with ‘’as shown in the Figure 2 for the case Sc4N4, the most suitable functional’’. Thus, the caption of Figure 2 refers to same as in the text.

Comment 2

‘’ In addition, in this figure, it can be seen that the B3LYP peak matches well with the EOM-CCSD peak at ~2.8 eV, while PBE underestimates excitation energy of this peak evidently. I really don't know how the authors drew the conclusion that PBE is the most suitable functional for estimating absorption spectrum of ScxNx. This point should be better clarified.’’

 Response

We thank the Reviewer for the comment and for giving us the opportunity to clarify it. It is fact, as shown in Figure 2, that PBE and B3LYP functionals are the best in comparison with the EOM_CCSD results.  The optical gap calculated with ΕΟΜ_CCDD is, indeed, almost the same as the corresponding one calculated with the B3LYP functional, while it differs by 8% from the corresponding gap calculated with PBE functional.  However, as we also notice from the Fig.2, the excitation energies calculated with PBE functional (compared to those calculated with B3LYP functional) are in better agreement with the corresponding energies calculated with EOM_CCDT. In conclusion, for this reason the PBE functional is chosen for the approximation and calculation of the absorption spectrum. Also, if the Reviewer considers that the above comment will help the understanding of the manuscript, we can add it to the revised text.

Round 2

Reviewer 2 Report

No, I cannot accept that you choose different xc-functionals for the different families just by looking at these plots. This is really arbitrary and not appropriate for investigations in physics, which can be (and should be) an exact science. An adequate method for comparing functions of a shape as your absorbance spectra (consisting of distinct peaks) is to first integrate them and then compare the integrals (in the sense of an L^2-norm of their difference). Thus, it is sensitive to the positions and heights of the peaks, while doing the L^2-norm directly (without integrating first) would be indifferent if the peaks are not so close to overlap. But this would be only for the Sc_4Y_4 structures, and you do not give any argument (or present results) as to why a given functional Z should be better for all structures for given Y if it is better for the Sc_4Y_4 structure. What I see is that the PBE functional is consistently (for all three families) the lowest in energy, then comes B3LYP, and then PBE0. In any case, since I probably cannot get you to redo your calculations with a common functional, I can only ask you to show in an objective way (as I have sketched) that your chosen functionals are best for the respective families (and not only for one representative, but show that it is transferable), or delete this whole section and just state that you have used different functionals for the different families. But then you would have to refrain from any comparisons between the families such as in the conclusions.

You write that except where mentioned, the structures do not have imaginary frequencies, but despite my explicit question, you do not specify whether you have allowed symmetry breaking. Thus I would like to have the force constant matrices to be able to verify myself: please provide them (together with a list of coordinates of the atoms) to me, say for the Cross configuration of Fig. 11h and Cross 5 of Fig. 11m, for Sc_nN_n.

Minor editing of English language required

Author Response

We would like to thank the Reviewer for his/her comments that helped us to significantly improve the quality of the manuscript.

Comment 1

''In any case, since I probably cannot get you to redo your calculations with a common functional, I can only ask you to show in an objective way (as I have sketched) that your chosen functionals are best for the respective families (and not only for one representative, but show that it is transferable), or delete this whole section and just state that you have used different functionals for the different families. But then you would have to refrain from any comparisons between the families such as in the conclusions''.

Response 1

We totally agree with the Reviewer and we deleted Figures 2 - 4 (as well as their comments). Also, we avoided comparisons between studied families.

Comment 2

''You write that except where mentioned, the structures do not have imaginary frequencies, but despite my explicit question, you do not specify whether you have allowed symmetry breaking''.

Response 2

Thank you for pointing this out. We did allow symmetry breaking in our calculations.

Comment 3

''Thus I would like to have the force constant matrices to be able to verify myself: please provide them (together with a list of coordinates of the atoms) to me, say for the Cross configuration of Fig. 11h and Cross 5 of Fig. 11m, for Sc_nN_n''.

Response 3

We have therefore clarified this point and are sending you the requested files as attachments to the journal assistant editor's email. (We could not find way to send them directly to you).  

Round 3

Reviewer 2 Report

Yes, the authors have adequately responded to my comments and modified the manuscript accordingly, and the claim about the stability of the considered structures, which I doubted originally, seems valid according to the provided interaction matrices. Also the symmetry is obviously broken. Thus, the manuscript can be published as it is.

I checked “Minor editing required”: To give examples, I mention the first line of the abstract, where the definite article before “Density Functional Theory” should be dropped, or also the last sentence of the abstract,  where “Those” should rather be “These”

Author Response

Thank you very much for your comments.

Indeed, there were grammatical errors, which were corrected in the revised manuscript.